# Complete dissection of transcription elongation reveals slow translocation of RNA polymerase II in a linear ratchet mechanism

Manchuta Dangkulwanich[1,2†], Toyotaka Ishibashi[1,3†], Shixin Liu[1,4†], Maria L Kireeva[5], Lucyna Lubkowska[5], Mikhail Kashlev[5], Carlos J Bustamante[1,2,3,4,6,7]*

[1]Jason L Choy Laboratory of Single-Molecule Biophysics, University of California, Berkeley, Berkeley, United States; [2]Department of Chemistry, University of California, Berkeley, Berkeley, United States; [3]California Institute for Quantitative Biosciences, University of California, Berkeley, Berkeley, United States; [4]Department of Physics, Howard Hughes Medical Institute, University of California, Berkeley, Berkeley, United States; [5]Gene Regulation and Chromosome Biology Laboratory, Center for Cancer Research–National Cancer Institute, Frederick, United States; [6]Department of Molecular and Cell Biology, University of California, Berkeley, Berkeley, United States; [7]Physical Biosciences Division, Lawrence Berkeley National Laboratory, Berkeley, United States

*For correspondence: carlos@alice.berkeley.edu

†These authors contributed equally to this work

Competing interests: The authors declare that no competing interests exist.

**Abstract** During transcription elongation, RNA polymerase has been assumed to attain equilibrium between pre- and post-translocated states rapidly relative to the subsequent catalysis. Under this assumption, recent single-molecule studies proposed a branched Brownian ratchet mechanism that necessitates a putative secondary nucleotide binding site on the enzyme. By challenging individual yeast RNA polymerase II with a nucleosomal barrier, we separately measured the forward and reverse translocation rates. Surprisingly, we found that the forward translocation rate is comparable to the catalysis rate. This finding reveals a linear, non-branched ratchet mechanism for the nucleotide addition cycle in which translocation is one of the rate-limiting steps. We further determined all the major on- and off-pathway kinetic parameters in the elongation cycle. The resulting translocation energy landscape shows that the off-pathway states are favored thermodynamically but not kinetically over the on-pathway states, conferring the enzyme its propensity to pause and furnishing the physical basis for transcriptional regulation.

## Introduction

Transcription constitutes the first and a central regulatory step for gene expression (*Greive and von Hippel, 2005*; *Coulon et al., 2013*). During the process of RNA synthesis, RNA polymerase (RNAP) converts the energy from chemical catalysis of the nucleoside triphosphate (NTP) into mechanical translocation along the DNA template. Two classes of mechanisms have been offered to describe the mechanochemical coupling of transcription elongation. The first class, known as the 'power stroke' mechanism, suggests that the forward translocation of RNAP is directly driven by a chemical step such as the release of the pyrophosphate ($PP_i$) (*Yin and Steitz, 2004*). The second class, known as the 'Brownian ratchet' mechanism, postulates that the polymerase oscillates back and forth on the DNA template between a pre- and a post-translocated state at the beginning of each nucleotide addition

**eLife digest** The production of a protein inside a cell starts with a region of the DNA inside the cell nucleus being transcribed to form a molecule of messenger RNA. This process involves an enzyme called RNA polymerase that moves along the DNA, reading the bases and making a complementary strand of messenger RNA from molecules called nucleoside triphosphates (NTPs). Just as there are four different bases in DNA, there are four different natural NTPs. In addition to supplying the correct bases for the messenger RNA molecule, these NTPs also provide the energy needed to drive the transcription process.

In many species the RNA polymerase oscillates between two neighbouring positions on the DNA, with this back-and-forth motion–which is powered by thermal energy–being converted into forward movement of the enzyme along the DNA when a new NTP binds to the growing messenger RNA molecule. It has long been assumed that the back-and-forth motion occurs much faster than the overall reaction of adding one NTP to the messenger RNA. This assumption has now been tested by using a single-molecule assay to monitor transcription in real time.

Dangkulwanich et al. measured the elongation velocities of yeast RNA polymerase II (Pol II) on bare DNA and on DNA in which a nucleosome–a structure that consists of a segment of DNA wrapped around histone proteins–had been placed as a "road block" in front of the enzyme. Surprisingly, the rate of the back-and-forth motion was found to be comparable in magnitude to the rate for adding one molecule of NTP. Dangkulwanich et al. also measured the rates associated with a process called backtracking in which the polymerase moves away from the transcription site to "pause" the process. These measurements show that there is a delicate balance between elongation and pausing during transcription.

Overall, by revealing the energy landscape associated with transcription, the work of Dangkulwanich et al. will bring us closer to the goal of creating a molecular movie of this extremely important–and complex–process.

cycle, and that such thermally-driven motions are rectified to the post-translocated state by the incorporation of the incoming NTP (*Guajardo and Sousa, 1997*). After extensive structural and biochemical investigations, it is now generally thought that multi-subunit RNAPs, including bacterial and eukaryotic enzymes, function through the Brownian ratchet mechanism (*Komissarova and Kashlev, 1997a*; *Bai et al., 2004*; *Bar-Nahum et al., 2005*; *Brueckner and Cramer, 2008*). This mechanism received further support from single-molecule studies, which followed the dynamics of individual transcription elongation complexes (TECs) (*Abbondanzieri et al., 2005*; *Bai et al., 2007*; *Larson et al., 2012*). Nonetheless, in order to explain the relationship between the elongation velocity and the external force applied to RNAP obtained from single-molecule experiments, the classical linear ratchet mechanism (*Figure 1*) had to be modified such that the incoming NTP must also bind to the pre-translocated TEC (*Figure 1—figure supplement 1*) (*Abbondanzieri et al., 2005*; *Larson et al., 2012*). In the pre-translocated TEC, the primary nucleotide binding site is occupied by the 3'-end of the nascent transcript. Thus, the branched Brownian ratchet scheme necessarily requires a secondary NTP binding site on the enzyme. However, the precise location of this secondary site and the mechanism by which the NTP is transferred to the primary site remain poorly defined.

Pausing is an off-pathway process that plays crucial roles in the regulation of transcription elongation (*Landick, 2006*; *Nudler, 2012*). In one view of the mechanisms of transcriptional pausing, RNAP first enters an elemental pause state (*Herbert et al., 2006*; *Toulokhonov et al., 2007*; *Sydow et al., 2009*), whose structural evidence was recently presented in bacterial RNAP (*Weixlbaumer et al., 2013*). However, similar evidence is lacking for eukaryotic polymerases. These elemental pauses can be subsequently stabilized into longer-lived pauses by the formation of a hairpin structure in the nascent RNA transcript or by RNAP backtracking (*Artsimovitch and Landick, 2000*; *Herbert et al., 2008*). The backtracking process is caused by upstream movements of the polymerase, displacing the 3'-end of the nascent RNA away from the active site into the secondary channel of the enzyme (*Nudler et al., 1997*; *Komissarova and Kashlev, 1997b*). An alternative view poses that most pauses are attributed to backtracking, which can be described as a one-dimensional random walk of the enzyme along the DNA template (*Galburt et al., 2007*; *Mejia et al., 2008*; *Depken et al., 2009*;

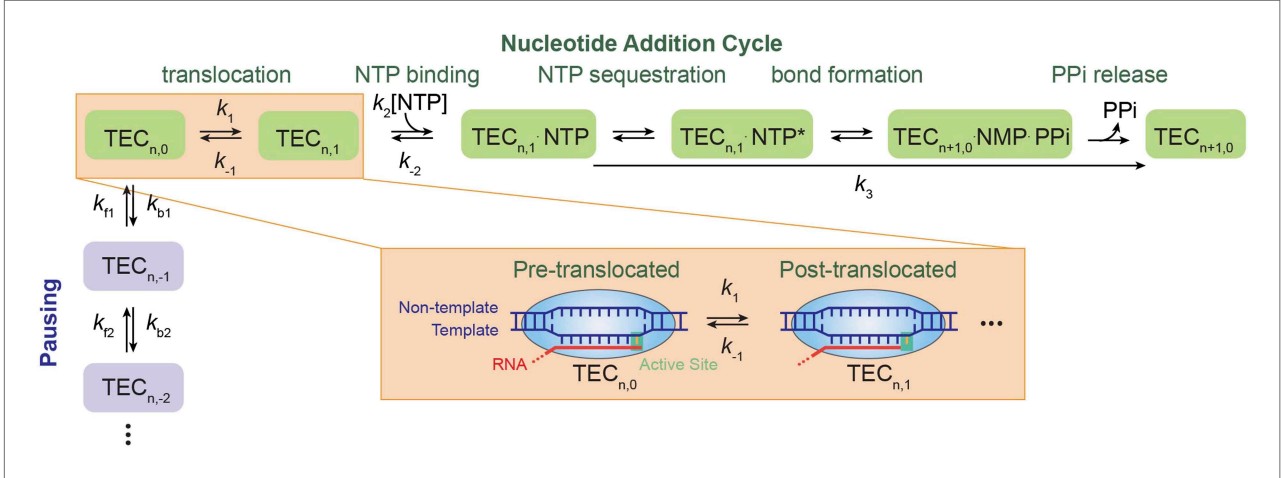

**Figure 1**. Nucleotide addition cycle and off-pathway pausing of transcription elongation. The nucleotide addition phase and the pausing phase are colored in green and blue, respectively. At the beginning of a nucleotide addition cycle, the transcription elongation complex (TEC) with a transcript length of $n$ thermally fluctuates between the pre-translocated state ($TEC_{n,0}$) and the post-translocated state ($TEC_{n,1}$) with a forward rate constant $k_1$ and a reverse rate constant $k_{-1}$. After translocation, the incoming NTP binds to the active site with a binding rate constant $k_2$ and a dissociation rate constant $k_{-2}$. NTP binding is followed by NTP sequestration, bond formation, and $PP_i$ release, which are collectively described by a single catalysis rate constant $k_3$ in our study. Upon the release of the $PP_i$, TEC is reset to the pre-translocated state ($TEC_{n+1,0}$) and ready for the next nucleotide addition cycle. From the pre-translocated state, the polymerase can also enter the off-pathway pausing phase by backtracking. The pausing kinetics are determined by the backward stepping rate constants $k_{bn}$ and forward stepping rate constants $k_{fn}$. The inset shows cartoon configurations of the TEC in a pre-translocated and a post-translocated state.

The following figure supplements are available for figure 1:

**Figure supplement 1**. A branched Brownian ratchet model for the nucleotide addition cycle.

*Hodges et al., 2009*). RNA synthesis resumes when the polymerase diffusively realigns its active site with the 3'-end of the transcript.

Both the nucleotide addition phase and the pausing phase are closely regulated by conserved structural motifs near the active center of the polymerase, namely the bridge helix and the trigger loop (TL) (*Bar-Nahum et al., 2005*; *Wang et al., 2006*; *Vassylyev et al., 2007*; *Brueckner and Cramer, 2008*; *Kaplan et al., 2008*; *Tan et al., 2008*). In order to understand the mechanism of transcription and its regulation, it is important to achieve a detailed description of both on- and off-pathway kinetics of the elongation reaction. Previous efforts to dissect the kinetic scheme of transcription elongation have assumed that the forward and reverse translocation steps of the Brownian ratchet occur in rapid equilibrium relative to the chemical steps in the nucleotide addition cycle (*Guajardo and Sousa, 1997*; *Bai et al., 2004*; *Abbondanzieri et al., 2005*; *Tadigotla et al., 2006*). However, the assumption of fast translocation equilibrium has never been experimentally validated. In fact, recent studies suggested that the translocation step may be partially rate-limiting for the nucleotide addition cycle, which gives rise to the heterogeneous elongation rates at different template positions (*Nedialkov et al., 2003*; *Kireeva et al., 2010*; *Maoiléidigh et al., 2011*; *Malinen et al., 2012*; *Nedialkov et al., 2012*; *Imashimizu et al., 2013*).

In this work, we sought to achieve a comprehensive kinetic characterization of transcription elongation without making any assumption about the rate-limiting mechanism of the reaction. We used an optical tweezers assay to follow the transcription trajectories of single yeast RNA polymerase II (Pol II) molecules under a variety of conditions, including varying NTP concentrations, assisting and opposing applied forces, and different tracks (bare and nucleosomal DNA). In vivo, eukaryotic DNA is organized around histone octamers to form nucleosomes, which impose physical barriers to transcription elongation. We have previously demonstrated that a transcribing Pol II cannot actively unravel a wrapped nucleosome. Instead, the polymerase pauses and waits until the local nucleosomal DNA spontaneously unwraps and permits Pol II to advance (*Hodges et al., 2009*; *Bintu et al., 2012*). Here we used the nucleosomal barrier as a tool to specifically perturb forward translocation of a transcribing Pol II and

separately measured the forward and reverse translocation rates. Surprisingly, we found that the forward translocation rate is of the same order of magnitude as the catalysis rate, in contradiction to previous assumptions of fast translocation. This finding reveals that translocation and catalysis together constitute the rate-limiting steps in the nucleotide addition cycle. As a consequence, we were able to rationalize the observed force–velocity relationship of the enzyme with a linear Brownian ratchet scheme in which the incoming NTP only binds to the post-translocated TEC, thus reconciling bulk and single-molecule data and arriving at a unifying view of the transcription elongation process. We further obtained all the major kinetic parameters in the nucleotide addition phase and the pausing phase of the elongation cycle. The energy landscape for transcription elongation derived from these parameters shows that: (i) the enzyme thermodynamically favors the pre-translocated state to the post-translocated state; (ii) entry into the 1-basepair (bp) backtracked state is easier than into further backtracked states; and (iii) from the pre-translocated state, the enzyme thermodynamically favors the backtracked states, but kinetically favors forward translocation. We also applied this analysis to a TL mutant Pol II, Rpb1-*E1103G* (*Malagon et al., 2006*), to quantitatively elucidate the roles of the TL in transcription elongation. Our results indicate that the conformational transitions of the TL control enzyme translocation, catalysis, and pausing, rendering it a vital target element for transcriptional regulation.

## Results

### NTP concentration dependence of elongation velocity and pausing frequency

The Brownian ratchet kinetic scheme for the nucleotide addition cycle of transcription elongation (*Figure 1*) can be simplified to:

$$\text{TEC}_{n,0} \underset{k_{-1}}{\overset{k_1}{\rightleftharpoons}} \text{TEC}_{n,1} \underset{k_{-2}}{\overset{k_2[NTP]}{\rightleftharpoons}} \text{TEC}_{n,1\cdot NTP} \overset{k_3}{\longrightarrow} \text{TEC}_{n+1,0}$$

where $k_1$ and $k_{-1}$ are the forward and reverse translocation rate constants, $k_2$ and $k_{-2}$ are the NTP binding and dissociation rate constants, and $k_3$ is the combined catalysis rate constant that includes NTP sequestration, bond formation, and $PP_i$ release. Because of the large equilibrium constant of transcription elongation and the very low $PP_i$ concentration (1 µM) in the buffer, $k_3$ was considered essentially irreversible (*Erie et al., 1992*). Using the concept of net rate constants (*Cleland, 1975*), we can replace the reversible rate constants between two adjacent states with a single net rate constant and re-write the above scheme as:

$$\text{TEC}_{n,0} \overset{k_1^{net}}{\longrightarrow} \text{TEC}_{n,1} \overset{k_2^{net}}{\longrightarrow} \text{TEC}_{n,1\cdot NTP} \overset{k_3}{\longrightarrow} \text{TEC}_{n+1,0}$$

$k_1^{net}$ and $k_2^{net}$ are the net rate constants for translocation and NTP binding, respectively, which are given by:

$$k_2^{net} = k_2[NTP] \cdot \frac{k_3}{k_{-2} + k_3} \qquad (1)$$

$$k_1^{net} = k_1 \cdot \frac{k_2^{net}}{k_{-1} + k_2^{net}} = \frac{k_1 k_2 k_3 [NTP]}{k_{-1}(k_{-2} + k_3) + k_2 k_3 [NTP]} \qquad (2)$$

The time the enzyme takes to finish one nucleotide addition cycle ($\tau$) equals the step size of the polymerase ($d = 1$ nt) divided by the pause-free velocity ($v$), and also equals the sum of the inverse of each net rate:

$$\tau = \frac{d}{v} = \frac{1}{k_1^{net}} + \frac{1}{k_2^{net}} + \frac{1}{k_3} \qquad (3)$$

Plugging *Equations 1* and *2* into *Equation 3* yields the following expression for the pause-free velocity:

$$v = \frac{k_1 k_3}{k_1 + k_3} \cdot \frac{[NTP]}{\frac{(k_1 + k_{-1}) \cdot (k_{-2} + k_3)}{(k_1 + k_3) \cdot k_2} + [NTP]} \cdot d \qquad (4)$$

We note that this expression is more general than those shown in previous studies (*Abbondanzieri et al., 2005*; *Bai et al., 2007*), as it is derived without assuming local equilibration of translocation and NTP binding. In particular, we describe the kinetics of the translocation step with $k_1$ and $k_{-1}$, instead of a single equilibrium constant $K_\delta = k_{-1}/k_1$. Such treatment is a prerequisite to explicitly determine the forward and reverse translocation rates. *Equation 4* can be simplified to the Michaelis–Menten equation form:

$$v = \frac{V_{max}[NTP]}{K_M + [NTP]} \qquad (5)$$

where $V_{max} = \frac{k_1 k_3}{k_1 + k_3} \cdot d$, and $K_M = \frac{k_1 + k_{-1}}{k_1 + k_3} \cdot \frac{k_{-2} + k_3}{k_2}$.

We followed the transcriptional dynamics of individual Pol II molecules with a dual-trap optical tweezers instrument. One laser trap holds a polystyrene bead attached to a stalled Pol II molecule, while the other trap holds another bead attached to the upstream DNA template (assisting force geometry; *Figure 2A*) or to the downstream template (opposing force geometry, not shown). Upon addition of NTP, transcription restarts, resulting in a change of the DNA tether length and thereby a variation of the force applied to Pol II. Single-molecule transcription trajectories were collected at a range of NTP concentrations (35 μM–2 mM) (*Figure 2B,C*). The relationship between pause-free velocity (*v*) and [NTP] for the wild-type enzyme fits well to *Equation 5*, with $V_{max} = 25 \pm 3$ nt/s and $K_M = 39 \pm 12$ μM (errors are SEM) (*Figure 2D*, gray line). We also examined the dynamics of the E1103G mutant Pol II, which is known to transcribe DNA at a faster overall velocity than the wild-type (*Figure 2B,C*) (*Malagon et al., 2006*; *Kireeva et al., 2008*). We found that the maximum pause-free velocity of the mutant is ~1.5-fold higher than that of the wild-type, with $V_{max} = 38 \pm 5$ nt/s and $K_M = 62 \pm 15$ μM (*Figure 2D*, blue line).

As shown in the example trajectories (*Figure 2B,C*), transcription elongation is punctuated by pauses of various durations. Pause density, $\rho_{pause}$, is defined as the average number of pauses per bp of template transcribed. As the concentration of NTP goes up, the pause-free velocity increases and the apparent $\rho_{pause}$, which counts pauses lasting longer than 1 s, decreases (*Figure 2E*). The same trend was also observed for the mutant Pol II (*Figure 2F*). The inverse relationship between *v* and $\rho_{pause}$ indicates that elongation and pausing are in kinetic competition and that pausing occurs prior to NTP binding (*Artsimovitch and Landick, 2000*; *Davenport et al., 2000*; *Forde et al., 2002*; *Landick, 2006*; *Mejia et al., 2008*). Note that pausing has also been observed to occur after NTP binding at certain sequences for *Escherichia coli* RNAP; however, yeast Pol II does not seem to employ such mechanism (*Kireeva and Kashlev, 2009*). The pause-free velocities and apparent pause densities at various NTP concentrations are summarized in *Table 1*.

## Determine the stepping rates during a backtracked pause

Backtracking is a major mechanism for transcriptional pauses. We have previously modeled backtracking as a one-dimensional random walk of the enzyme along the DNA template (*Hodges et al., 2009*). In this model, Pol II diffuses back and forth on DNA with a forward stepping rate constant $k_f$ and a backward stepping rate constant $k_b$ during a backtracked pause. These rate constants are dependent on the applied force (*F*, which is positive for assisting forces and negative for opposing forces) according to:

$$k_f = k_0 e^{F \cdot \Delta / k_B T} \qquad (6)$$

$$k_b = k_0 e^{-F \cdot (1 - \Delta) / k_B T} \qquad (7)$$

where $k_0$ is the intrinsic zero-force stepping rate constant of Pol II diffusing along DNA during backtracking, $\Delta$ is the distance to the transition state for each step (taken to be 0.5 bp, or 0.17 nm), $k_B$ is the Boltzmann constant, and *T* is the temperature ($k_B T = 4.11$ pN·nm at 25°C). The probability density of pause durations, $\psi(t)$, is equivalent to the distribution of first-passage times for a particle diffusing on a one-dimensional lattice to return to the origin (*Depken et al., 2009*), and is given by:

$$\psi(t) = \sqrt{\frac{k_f}{k_b}} \frac{\exp[-(k_f + k_b)t]}{t} I_1\left(2t\sqrt{k_f k_b}\right) \qquad (8)$$

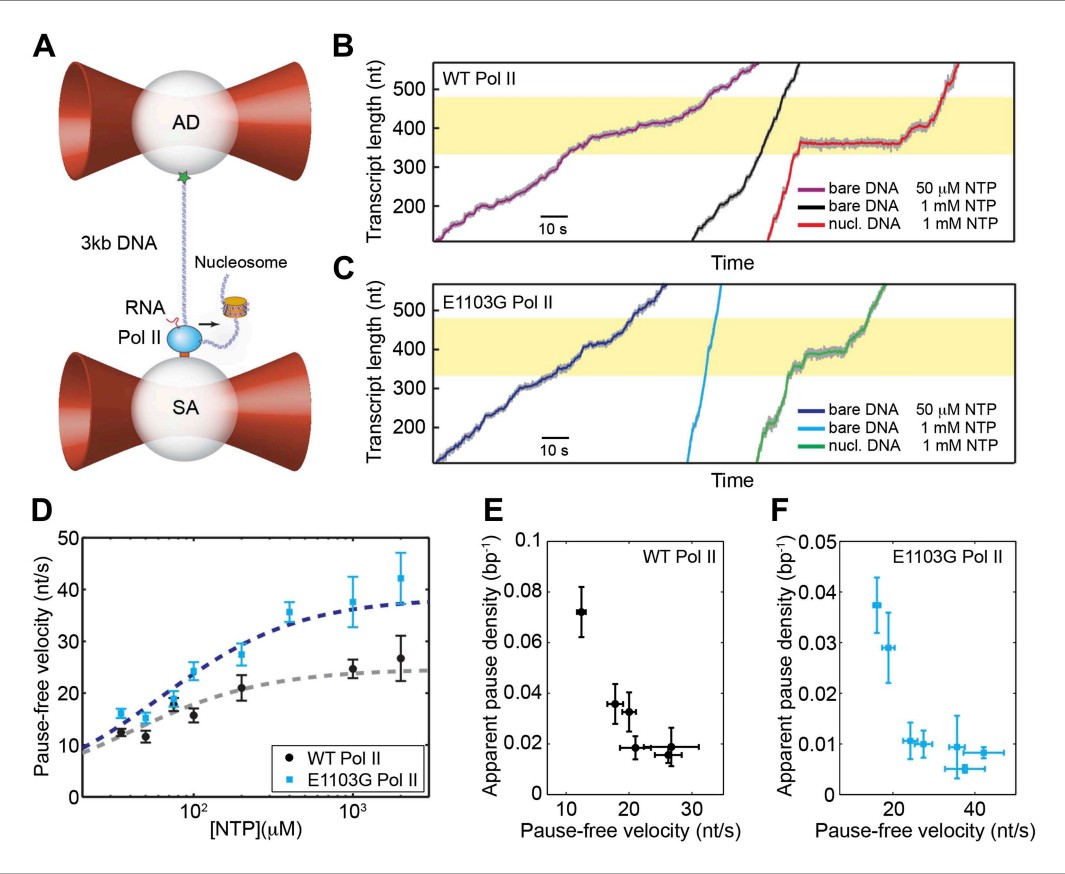

**Figure 2**. Single-molecule transcription assay. (**A**) Experimental setup for the single-molecule transcription assay. Each of the two optical traps holds a 2.1-µm polystyrene bead. Biotinylated Pol II is attached to the streptavidin (SA) bead. The upstream DNA is attached to the antibody (AD) bead via the digoxigenin–antidigoxigenin linkage. The black arrow indicates the direction of transcription. A nucleosome can be loaded on the downstream DNA as shown. (**B**) Example transcription trajectories of the wild-type Pol II at 50 µM NTP on bare DNA, 1 mM NTP on bare DNA, and 1 mM NTP in the presence of a nucleosome. The nucleosome positioning sequence (NPS) is represented by the yellow shaded region. (**C**) Example transcription trajectories of the E1103G mutant Pol II under various conditions. (**D**) Pause-free velocities of the wild-type (black) and mutant Pol II (blue) at various NTP concentrations. Dashed lines are fits to the Michaelis–Menten equation (***Equation 3***; $R^2$ = 0.80 for the wild-type; $R^2$ = 0.85 for mutant). (**E**) The apparent pause densities ($\rho_{pause}$) of the wild-type Pol II at different NTP concentrations are plotted against the corresponding pause-free velocities ($v$). (**F**) $\rho_{pause}$–$v$ relationship for the mutant enzyme. Error bars represent standard error of the mean (SEM).

The following figure supplements are available for figure 2:

**Figure supplement 1**. Cumulative distribution of the pause durations for the wild-type Pol II on bare DNA (black solid line) and nucleosomal DNA (red solid line).

**Figure supplement 2**. A gel-based time-coursed transcription assay of the wild-type Pol II on bare and nucleosomal DNA.

**Figure supplement 3**. A gel-based transcription assay of the wild-type Pol II and the E1103G mutant Pol II in various KCl concentrations.

**Figure supplement 4**. Mean dwell times of the wild-type Pol II at different nucleotide positions.

**Figure supplement 5**. Mean dwell times of the mutant Pol II at different nucleotide positions.

**Table 1.** Summary of pause-free velocities and apparent pause densities measured at various NTP concentrations

| Pol II | [NTP] (μM) | *N* | Pause-free velocity (nt/s) | Apparent pause density (bp⁻¹) |
|---|---|---|---|---|
| wild-type | 35 | 10 | 12.4 ± 0.7 | 0.0721 ± 0.0099 |
| | 50 | 11 | 11.6 ± 1.1 | 0.0526 ± 0.0086 |
| | 75 | 9 | 17.8 ± 1.2 | 0.0358 ± 0.0079 |
| | 100 | 13 | 15.7 ± 1.4 | 0.0326 ± 0.0077 |
| | 200 | 17 | 21.0 ± 2.4 | 0.0184 ± 0.0046 |
| | 1000 | 44 | 24.7 ± 1.8 | 0.0156 ± 0.0031 |
| | 2000 | 9 | 26.7 ± 4.3 | 0.0188 ± 0.0076 |
| E1103G | 35 | 10 | 16.1 ± 0.9 | 0.0374 ± 0.0055 |
| | 50 | 13 | 15.2 ± 1.0 | 0.0266 ± 0.0055 |
| | 75 | 13 | 18.9 ± 1.5 | 0.0290 ± 0.0069 |
| | 100 | 13 | 24.2 ± 1.7 | 0.0106 ± 0.0036 |
| | 200 | 13 | 27.4 ± 3.9 | 0.0100 ± 0.0027 |
| | 400 | 10 | 35.6 ± 1.9 | 0.0094 ± 0.0062 |
| | 1000 | 96 | 37.6 ± 4.9 | 0.0051 ± 0.0008 |
| | 2000 | 15 | 42.1 ± 4.9 | 0.0083 ± 0.0011 |

Data are shown as mean ± SEM. The apparent pause densities are determined by counting pauses that last between 1 s and 120 s. *N* is the number of single-molecule transcription trajectories at each condition.

where $I_1$ is the modified Bessel function of the first kind. We fit the distribution of pause durations for the wild-type enzyme on bare DNA to this model and extracted a characteristic $k_0$ of 1.3 ± 0.3 s⁻¹ (*Figure 2—figure supplement 1*, gray dashed line). Using the values of $k_0$ and the applied force in our experiment (6.5 pN), we calculated $k_f$ and $k_b$ to be 1.7 ± 0.4 s⁻¹ and 1.0 ± 0.3 s⁻¹, respectively (*Equations 6* and *7*).

## Pausing properties on nucleosomal DNA

Next, we investigated the transcriptional dynamics of Pol II through the nucleosome by loading a histone octamer on the 601 nucleosome positioning sequence (NPS) (*Lowary and Widom, 1998*) (*Figure 2B,C*, *Figure 2—figure supplements 2–5*). The wild-type enzyme displays a two-fold increase in the apparent pause density upon encountering the nucleosome (*Table 2*). The mean pause duration on nucleosomal DNA is significantly longer than that on bare DNA (*Table 2*; *Figure 2—figure supplement 1*). Similarly, the mutant Pol II displays higher pause density and longer pause duration in the presence of a nucleosome (*Table 2*).

It has been shown that the nucleosomal DNA can spontaneously unwrap and rewrap around the histones (*Li et al., 2005*; *Koopmans et al., 2008*; *Voltz et al., 2012*). The increased pause duration of

**Table 2.** Apparent pause densities and mean pause durations on bare DNA and nucleosomal DNA in the extended NPS region

| Pol II | DNA template | *N* | Apparent pause density (bp⁻¹) | Mean pause duration (s) |
|---|---|---|---|---|
| wild-type | Bare | 38 | 0.0153 ± 0.0041 | 3.9 ± 0.6 |
| | Nucleosomal | 94 | 0.0280 ± 0.0036 | 9.4 ± 0.8 |
| E1103G | Bare | 85 | 0.0046 ± 0.0015 | 3.9 ± 0.5 |
| | Nucleosomal | 64 | 0.0202 ± 0.0050 | 7.6 ± 1.0 |

Data are shown as mean ± SEM. The extended NPS region spans −115 nt to+85 nt relative to the nucleosomal dyad.

Pol II on nucleosomal DNA can be explained by rewrapping of the DNA downstream of a backtracked Pol II, which prevents the polymerase from diffusing back to the 3'-end of the nascent RNA to resume transcription (*Hodges et al., 2009*; *Bintu et al., 2012*). Because one bp of nucleosomal DNA fluctuates much faster (>1000 s$^{-1}$; see 'Materials and methods' for the derivation) than Pol II stepping (~1 s$^{-1}$), the nucleosomal DNA in front of the polymerase reaches wrapping/unwrapping equilibrium between each backtracking step. It follows that the pause durations on nucleosomal DNA can be drawn from the same distribution as on bare DNA, except that the effective forward stepping rate is reduced by a factor, $\gamma_u$, corresponding to the fraction of time the local nucleosomal DNA is unwrapped (*Hodges et al., 2009*), that is $k_{f(nucl)} = \gamma_u \cdot k_f$. The backward stepping rate $k_b$ is not affected by the nucleosome, because little histone transfer occurs in our experimental geometry where the DNA template is under tension (*Hodges et al., 2009*; *Bintu et al., 2011*) and therefore the polymerase does not encounter any roadblock when it diffuses backward. The distribution of pause durations for wild-type Pol II on nucleosomal DNA can be correctly fit by this model with a $\gamma_u$ value of 0.6 ± 0.2 (*Figure 2—figure supplement 1*, red dashed line).

## Determine the rates of forward translocation and catalysis by comparing pause-free velocities on nucleosomal DNA and bare DNA

Having understood the effect of the nucleosomal barrier on the pausing dynamics, we then turned our attention to its effect on the on-pathway elongation kinetics. Interestingly, we found that the nucleosome also delays the transcribing enzyme by modulating its pause-free velocity. As the wild-type Pol II transcribes through nucleosomal DNA at saturating [NTP], its mean pause-free velocity decreases by 14% from 26.9 ± 0.8 nt/s to 23.2 ± 0.6 nt/s (*Figure 3A*). The mutant Pol II is even more dramatically slowed down by the nucleosome, with its mean pause-free velocity reduced by 35% from 39.8 ± 0.6 nt/s to 26.0 ± 0.7 nt/s (*Figure 3B*).

We have previously shown that a transcribing Pol II cannot actively open a wrapped nucleosome; instead, the enzyme passively waits for the DNA immediately in front of it to spontaneously unwrap and then translocates forward through a locally unwrapped nucleosome (*Hodges et al., 2009*). Since the fluctuations of local nucleosomal DNA occur orders of magnitude faster than the translocations of Pol II during backtracking, we assume that they are also much faster than the on-pathway translocation steps of Pol II. Under this assumption, local DNA reaches wrapping/unwrapping equilibrium before

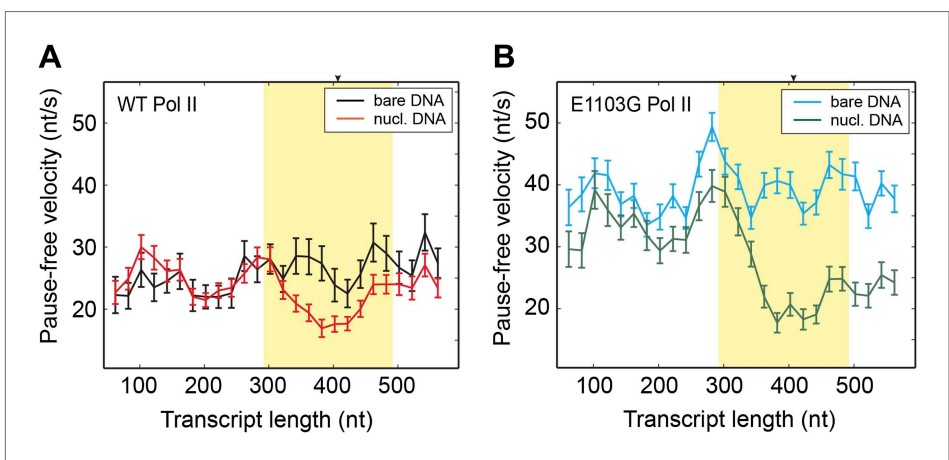

**Figure 3**. Comparison of pause-free velocities on bare DNA and nucleosomal DNA. (**A**) Pause-free velocities of the wild-type Pol II on bare DNA (black) and nucleosomal DNA (red) are plotted as a function of the transcript length. The nucleosomal dyad position corresponds to a transcript length of 407 nt. The extended NPS region (−115 nt to +85 nt relative to the nucleosomal dyad) is highlighted in yellow. The arrow on the top axis marks the position of the dyad. (**B**) Pause-free velocities of the E1103G mutant Pol II on bare DNA (blue) and nucleosomal DNA (green) are plotted as a function of the transcript length. These experiments were conducted at 1 mM NTP. Note that after the polymerase exits the nucleosome, the velocity does not return to the same level of that on bare DNA. This observation could be rationalized if the nucleosome rolls along the DNA and remains ahead of the transcribing polymerase in a fraction of the traces. Error bars are SEM.

Pol II makes a translocation step and the forward translocation rate ($k_1$) is effectively reduced by the fraction of time the local nucleosomal DNA is unwrapped ($\gamma_u$). The reverse translocation rate ($k_{-1}$) is unlikely to be affected, again due to the lack of a roadblock against reverse translocation. Thus, according to *Equation 5*, the maximum pause-free velocity for nucleosomal DNA transcription is:

$$V_{\max(nucl)} = \frac{\gamma_u \cdot k_1 \cdot k_3}{(\gamma_u \cdot k_1) + k_3} \cdot d \qquad (9)$$

In comparison, the maximum pause-free velocity for bare DNA transcription is:

$$V_{\max} = \frac{k_1 k_3}{k_1 + k_3} \cdot d \qquad (10)$$

Using an optimal $\gamma_u$ value of 0.6, we solved *Equations 9* and *10* and obtained $k_1 = 112 \pm 30$ s$^{-1}$ (indeed much slower than local DNA wrapping/unwrapping) and $k_3 = 35 \pm 3$ s$^{-1}$ for the wild-type. Importantly, these values show that the forward translocation rate is only three times faster than the catalysis rate and, therefore, has a significant contribution to the overall elongation velocity. For the mutant Pol II, translocation becomes even slower than catalysis ($k_1 = 50 \pm 4$ s$^{-1}$ and $k_3 = 195 \pm 65$ s$^{-1}$). The mutant's higher $k_3$ compensates for its lower $k_1$, rendering its overall velocity faster than that of the wild type. We note that these numbers were extracted by using the average values of the pause-free velocity and $\gamma_u$ over the whole nucleosomal region. Such a simplifying treatment is based on the observations that both the pause-free velocity (*Figure 3*) and the local DNA wrapping equilibrium (*Bintu et al., 2012*) do not change substantially along the NPS.

## The first backtracking step is distinct from subsequent steps

The pause density, $\rho_{pause}$, is governed by the kinetic competition between pause entry and elongation. Previously, an overall elongation rate, which includes translocation, NTP binding, and catalysis, was used in the expression for $\rho_{pause}$ (*Herbert et al., 2006*; *Hodges et al., 2009*; *Zhou et al., 2011*). A more accurate treatment is to use the elementary rate constant in the elongation pathway directly connected to pausing, which is the net rate constant for forward translocation, $k_1^{net}$ (*Figure 1*; *Equation 2*):

$$\rho_{pause} = \frac{k_{b1}}{k_{b1} + k_1^{net}} = \frac{k_{b1}}{k_{b1} + \dfrac{[NTP]}{\dfrac{k_{-1}(k_{-2}+k_3)}{k_2 k_3} + [NTP]} \cdot k_1} \qquad (11)$$

where $k_{b1}$ is the rate constant of entering the 1-bp backtracked pausing state. At saturating NTP concentrations ($[NTP] \gg k_{-1}(k_{-2}+k_3)/(k_2 k_3)$), $k_1^{net}$ becomes equivalent to $k_1$. Hence

$$\rho_{pause(sat)} = \frac{k_{b1}}{k_{b1} + k_1} \qquad (12)$$

where $\rho_{pause(sat)}$ is the pause density at saturating NTP concentration. In order to obtain a true pause density, the apparent $\rho_{pause}$ needs to be corrected to include pauses shorter than 1 s that are missed by our pause detection algorithm. After such a correction ('Materials and methods'), the total $\rho_{pause(sat)}$ is 0.045 ± 0.012 bp$^{-1}$. Solving *Equation 12* yields $k_{b1} = 5.3 \pm 2.0$ s$^{-1}$. This value is approximately five times larger than subsequent backward stepping rates, which are force-biased stepping rates obtained from *Equation 7* ($k_{bn} = 1.0 \pm 0.3$ s$^{-1}$, n≥2). The difference between $k_{b1}$ and $k_{bn}$ indicates that the first backtracking transition is easier to make than subsequent backtracking transitions. Using this value of $k_{b1}$, along with the value of $\gamma_u$ obtained above, we can predict a nucleosomal pause density of 0.035 ± 0.015 bp$^{-1}$ for pauses longer than 1 s, which agrees with the experimental measurement (*Table 2*).

We then compared the pausing kinetics between the wild-type and the mutant enzymes. Interestingly, on bare DNA, the mutation only affects the distribution of pauses that are shorter than 2 s (*Figure 4A*, p=0.003, Kolmogorov-Smirnov test). In contrast, the distributions of longer pauses are indistinguishable between the mutant and the wild-type Pol II (*Figure 4—figure supplement 1*, p=0.9). It is possible to rationalize this observation if the mutation selectively influences the kinetics of

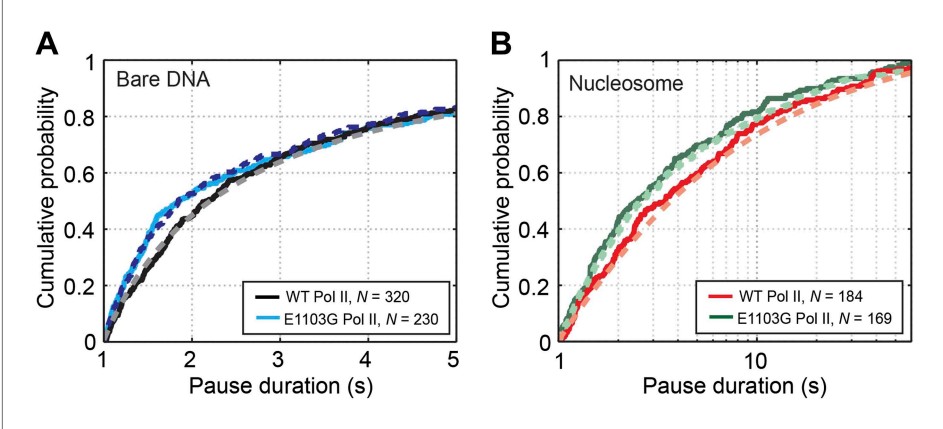

**Figure 4**. Pause durations on bare DNA and nucleosomal DNA. (**A**) Cumulative distributions of the pause durations on bare DNA for the wild-type Pol II (black solid line) and the mutant enzyme (blue solid line). The wild-type curve is fit to the one-dimensional random walk model for backtracked pausing (gray dashed line). The blue dashed line represents the simulated pause duration distribution for the mutant enzyme, using a $k_{f1}$ value of 4 s$^{-1}$. (**B**) Cumulative distributions of the pause durations in the nucleosome region for the wild-type enzyme (red solid line) and the mutant enzyme (green solid line). The wild-type curve is fit to the one-dimensional diffusion model for backtracked pausing, using a $\gamma_u$ value of 0.6 (red dashed line). The green dashed line is the simulated pause duration distribution for nucleosomal DNA transcription by the mutant enzyme, using a $k_{f1}$ value of 4 s$^{-1}$.

The following figure supplements are available for figure 4:

**Figure supplement 1**. Cumulative pause duration distributions of pauses longer than 3 s.

**Figure supplement 2**. Comparison between the experimentally obtained distribution of pause durations and the simulated distribution for the nucleosomal DNA transcription by the mutant Pol II.

the first backtracking step ($k_{b1}$ and/or $k_{f1}$) without affecting subsequent backtracking steps, given that pauses of short durations involve small backtracking excursions and that entering the 1-bp backtracked state is distinct from entering longer backtracked ones ($k_{b1}$ is different from $k_{bn}$, n≥2). The first backward stepping rate ($k_{b1}$) only influences the pause density but not the pause duration, while the first forward stepping rate ($k_{f1}$) does affect the pause duration. Specifically, the increase in short pauses can be explained if the mutation increases $k_{f1}$ and accelerates the return from a pause to active elongation. Indeed, Monte Carlo kinetic simulations show that setting $k_{f1}$ to be larger than 4 s$^{-1}$—2.4-fold higher than the wild-type value (1.7 ± 0.4 s$^{-1}$; **Equation 6**)—can reproduce the experimentally observed pause duration distributions for the mutant Pol II on bare DNA (**Figure 4A**, blue dashed line) and nucleosomal DNA (**Figure 4B**, green dashed line, and **Figure 4—figure supplement 2**). Moreover, by comparing the experimentally measured and simulated pause densities using different $k_{b1}$ values, we can set a lower bound for the mutant's $k_{b1}$ to be 2.8 s$^{-1}$.

Taken together, we have shown that the rate of entering the 1-bp backtracked state is higher than those of entering further backtracked states, and that the E1103G mutation modulates the transition kinetics between the 1-bp backtracked state and the pre-translocated state. Until now, $k_{f1}$ and $k_{b1}$ have been assumed to be identical with the other stepping rates during backtracking ($k_{fn}$ and $k_{bn}$, n≥2) (**Galburt et al., 2007**; **Hodges et al., 2009**; **Bintu et al., 2012**). Our data here suggest that the first backtracking step should be treated differently, consistent with published structural data (**Wang et al., 2009**; **Cheung and Cramer, 2011**) (see 'Discussion').

## Determine the rate of reverse translocation

We have determined the rates of forward translocation ($k_1$) and catalysis ($k_3$) in the nucleotide addition cycle and shown that they are comparable. What remains unknown is the reverse translocation rate $k_{-1}$, which may also affect the elongation velocity under sub-saturating NTP conditions (**Equation 4**).

To determine $k_{-1}$, we examined the pause densities measured at various NTP concentrations. *Equation 11* can be re-written as:

$$\rho_{\text{pause}} = \frac{k_{b1}}{k_{b1} + \dfrac{[NTP]}{\dfrac{k_{-1}K}{k_3} + [NTP]} \cdot k_1} \qquad (13)$$

where $K = (k_{-2}+k_3)/k_2$. The total $\rho_{\text{pause}}$ as a function of [NTP] fits well to *Equation 13* (*Figure 5A,B*). Using the values of $k_1$, $k_3$, and $k_{b1}$ determined above, we obtained $k_{-1}K$ equal to $(4.7 \pm 0.5) \times 10^3$ µM·s$^{-1}$ and $(2.5 \pm 0.4) \times 10^4$ µM·s$^{-1}$ for the wild-type and the mutant enzymes, respectively.

We then revisited the relationship between the pause-free velocity and [NTP] (*Figure 2D*), which follows Michaelis–Menten kinetics. According to *Equation 5*, the Michaelis constant $K_M$ is expressed as:

$$K_M = \frac{k_1+k_{-1}}{k_1+k_3} \cdot \frac{k_{-2}+k_3}{k_2} = \frac{k_1+k_{-1}}{k_1+k_3} \cdot K \qquad (14)$$

Plugging the values of $K_M$, $k_1$, $k_3$, and $k_{-1}K$ into *Equation 14* yields the values of $K$ and $k_{-1}$ for the wild-type Pol II: $K = 9.2$ µM and $k_{-1} = 510$ s$^{-1}$. We could further calculate the translocation equilibrium constant, $K_\delta$ = [pre-translocated]/[post-translocated] = $k_{-1}/k_1$ = 4.6. This result indicates that the enzyme favors the pre-translocated state to the post-translocated one, in agreement with most previous reports (*Bar-Nahum et al., 2005*; *Bai et al., 2007*; *Kireeva et al., 2008*; *Maoileidigh et al., 2011*). For the mutant enzyme, we could set an upper bound of $K$ to be 100 µM and a lower bound of $k_{-1}$ to be 210 s$^{-1}$ (*Figure 5—figure supplement 1*). Assuming that the mutant shares a similar $K$ value with the wild-type, we calculated $k_{-1}$ to be ~2700 s$^{-1}$ and $K_\delta$ to be ~54 for the mutant Pol II (see 'Materials and methods' for a discussion of this assumption).

## Force–velocity relationship

A central piece of evidence previously used to favor a branched kinetic scheme (*Figure 1—figure supplement 1*) over a simpler linear scheme (*Figure 1*) for the nucleotide addition cycle is the relationship between the pause-free velocity ($v$) and the applied force ($F$) (*Abbondanzieri et al., 2005*; *Larson et al., 2012*). However, in those studies, translocation was assumed to be in rapid equilibrium relative to catalysis. Having explicitly determined the translocation rates ($k_{\pm1}$) and found that the forward

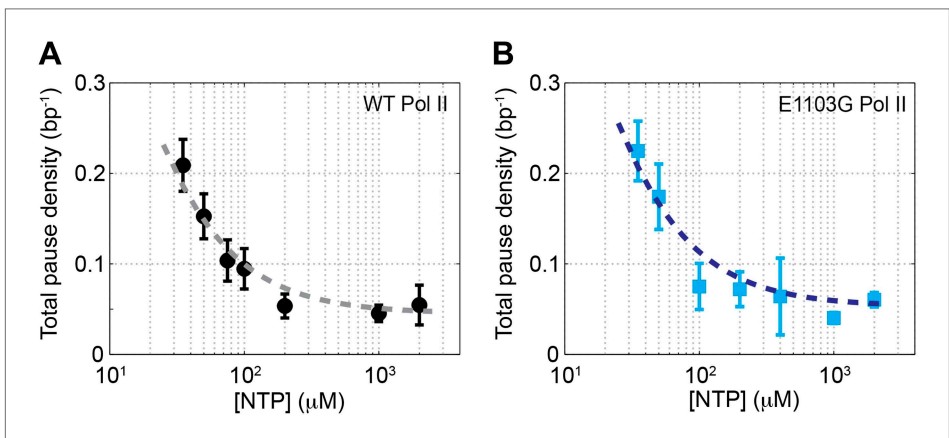

**Figure 5**. Relationship between pause density and NTP concentration. (**A**) The total pause density for the wild-type Pol II (black circles) is plotted against the NTP concentration. The gray dashed line is the fit to *Equation 13* (R² = 0.93). (**B**) $\rho_{\text{pause}}$–[NTP] relationship (blue squares) for the mutant Pol II is fit to *Equation 13* (blue dashed line, R² = 0.89).

The following figure supplements are available for figure 5:

**Figure supplement 1**. Constraining the value of $K$ for the mutant Pol II.

translocation rate ($k_1$) is comparable to the catalysis rate ($k_3$), we went on to examine whether a linear kinetic scheme (*Figure 1*) is sufficient to explain the *F–v* relationship, which for such scheme can be expressed as:

$$v(F) = \frac{k_1(F) \cdot k_3}{k_1(F) + k_3} \cdot \frac{[NTP]}{\dfrac{(k_1(F) + k_{-1}(F)) \cdot K}{k_1(F) + k_3} + [NTP]} \cdot d \qquad (15)$$

We assume that only the translocation transitions in the nucleotide addition cycle are force-sensitive and that the translocation rates depend on force according to the Boltzmann-type equation: $k_1(F) = k_1(0) \cdot e^{F\delta/k_B T}$, and $k_{-1}(F) = k_{-1}(0) \cdot e^{-F \cdot (1-\delta)/k_B T}$, where $\delta$ is the distance to the transition state for forward translocation, the only unknown variable left in *Equation 15*. We measured the pause-free velocity at different applied forces for both wild-type and mutant enzymes and obtained values in good agreement with previously published single-molecule data (*Larson et al., 2012*) (*Figure 6A,B*). The velocity of the wild-type enzyme shows a weak but detectable dependence on force, while the velocity of the mutant displays a much stronger force dependence. The *F–v* plots can be fit well to *Equation 15* with $\delta$ of 0.46 ± 0.09 bp for the wild-type (*Figure 6A*) and 0.24 ± 0.05 bp for the mutant (*Figure 6B*). Therefore, it is indeed possible to explain the observed force–velocity relationship of transcription elongation with a classic, non-branched Brownian ratchet mechanism, in which NTP binding occurs after translocation.

## Discussion

### Rate-limiting steps in the Brownian ratchet mechanism

RNAP transcribes DNA through a multi-step kinetic pathway. The rate-limiting nature of the various steps in the nucleotide addition cycle has so far remained largely conjectural. Almost all the existing kinetic studies of transcription elongation relied on the major assumption that translocation and NTP binding follow rapid equilibrium kinetics. As a result, the catalytic step occurring after NTP binding has been assigned to be rate-limiting of the overall elongation reaction.

The linear Brownian ratchet mechanism that assumes fast translocation equilibrium predicts that, as the NTP concentration increases, the force-sensitivity of the elongation velocity decreases and eventually vanishes, because the enzyme spends less time in the load-sensitive translocation steps.

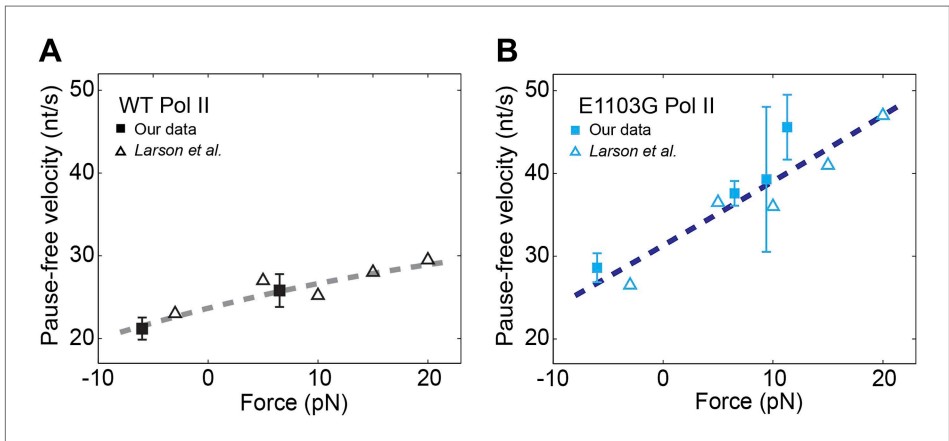

**Figure 6**. Relationship between transcription velocity and applied force. (**A**) The pause-free velocity of the wild-type Pol II is plotted against the applied force. Experimental data in the present study are shown in solid squares (error bars indicate SEM). Open triangles represent data from a previously published single-molecule study (*Larson et al., 2012*). The combined data are fit to the force-velocity relationship predicted by a linear Brownian ratchet model (dashed line), yielding a characteristic distance to the transition state $\delta$ = 0.46 ± 0.09 bp (error is SEM, $R^2$ = 0.88). Positive and negative force values indicate assisting and opposing forces, respectively. (**B**) The force-velocity relationship for the mutant Pol II. $\delta$ = 0.24 ± 0.05 bp for the mutant ($R^2$ = 0.85).

However, the *F–v* relationships of the enzyme obtained from optical tweezers studies have shown significant dependence of elongation velocity on external force even at saturating NTP concentrations (*Abbondanzieri et al., 2005*; *Bai et al., 2007*; *Larson et al., 2012*), in contradiction to the above prediction. To account for this discrepancy, a modified, branched ratchet model was proposed in which the NTP must also bind to a secondary site on the polymerase in the pre-translocated configuration. Although the existence of such additional binding site may be rationalized by the downstream allosteric site (*Holmes and Erie, 2003*; *Gong et al., 2005*), the 'E' site or pre-insertion site (*Westover et al., 2004*; *Temiakov et al., 2005*), or the tilted hybrid structure (*Cheung et al., 2011*), whether it constitutes a significant pathway in the elongation reaction and how it is related to the primary nucleotide binding pathway remain obscure. More importantly, the branched model neglects the possibility that the translocation steps may not be as fast as assumed.

In this study, we tested this possibility of slow translocation by placing a nucleosome in the path of the transcribing polymerase and directly determining the rates of forward and reverse translocation. Our analyses show that the forward translocation rate is in fact within the same order of magnitude as the catalysis rate. For the wild-type Pol II, $k_1$ is only 2.5 times higher than $k_3$ (*Figure 7A*; *Table 3*). For the E1103G mutant, $k_1$ even becomes the slowest step in the nucleotide addition cycle (*Table 3*). Hence, the translocation step is one of the rate-limiting transitions during transcription elongation.

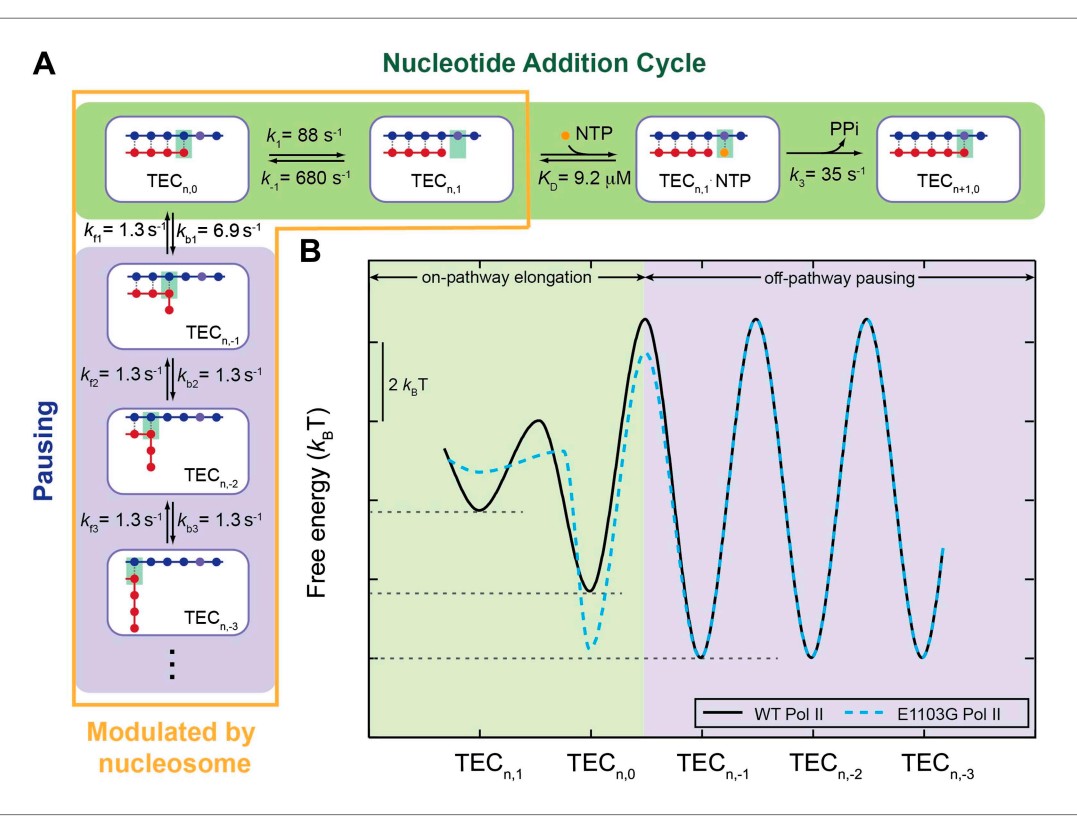

**Figure 7**. A quantitative kinetic model for transcription elongation. (**A**) A comprehensive kinetic characterization of the nucleotide addition phase (highlighted in green) and the pausing phase (highlighted in blue) for transcription by the wild-type Pol II. Inside the yellow box are the transitions affected by the nucleosomal barrier. (**B**) The schematic translocation free energy landscape at a given RNA length for the wild-type Pol II (solid black) and the E1103G Pol II (dashed cyan). The on-pathway elongation is highlighted in green and the off-pathway pausing is highlighted in blue.

The following figure supplements are available for figure 7:

**Figure supplement 1**. The schematic three-dimensional free energy landscape for transcription elongation by the wild-type Pol II at 1 mM NTP and zero force.

**Table 3.** Summary of kinetic parameters measured in this study

| Parameters | Wild-type Pol II | E1103G Pol II |
|---|---|---|
| $k_1$ (s$^{-1}$) | 88 ± 23 | *44 ± 4* |
| $k_{-1}$ (s$^{-1}$) | ~680 | *~4.1 × 10$^3$* |
| $K_\delta = k_{-1}/k_1$ | ~7.7 | *~92* |
| $K = (k_{-2}+k_3)/k_2$ (µM) | ~9.2 | ~9.2 |
| $k_3$ (s$^{-1}$) | 35 ± 3 | *195 ± 65* |
| $k_{b1}$ (s$^{-1}$) | 6.9 ± 2.6 | *~3.7** |
| $k_{f1}$ (s$^{-1}$) | 1.3 ± 0.3 | *~3.1** |
| $k_{bn}$ (s$^{-1}$), n ≥ 2 | 1.3 ± 0.3 | 1.3 ± 0.3 |
| $k_{fn}$ (s$^{-1}$), n ≥ 2 | 1.3 ± 0.3 | 1.3 ± 0.3 |

The values reported in the text were measured at 6.5 pN of applied assisting force and are normalized to zero force here. The italicized numbers indicate the parameters that are altered by the E1103G mutation. The asterisks indicate lower bounds of the corresponding values.

Translocation and catalysis together control the overall elongation velocity. These findings naturally explain the observed *F–v* relationship: because the enzyme always spends a considerable amount of time in the force-sensitive pre-translocated state even at high [NTP], we should always expect a force-dependence of the velocity. Moreover, a lower $k_1$ renders the velocity more sensitive to force, consistent with the experimental observation that the mutant Pol II shows a steeper *F–v* curve than the wild-type (***Figure 6A,B***). Therefore, our results demonstrate that a linear ratchet model can explain the transcriptional kinetics of Pol II and that it is not necessary to invoke a conceptually more complicated branched model, as long as the constraint of fast translocation equilibrium is relieved. Note that although our data argue against rapid oscillation of the ratchet, they still support the notion that the enzyme is able to spontaneously diffuse along the DNA between the pre- and post-translocated states, as suggested by the Brownian ratchet mechanism.

We extracted the values of $k_1$ and $k_3$ by comparing the maximum pause-free velocities on bare DNA and nucleosomal DNA (***Equations 9*** and ***10***). In principle, $k_1$ and $k_3$ can also be determined by examining $V_{max}$ as a function of applied force:

$$V_{max}(F) = \frac{k_1(F) \cdot k_3}{k_1(F) + k_3} \cdot d \tag{16}$$

where $k_1(F) = k_1(0) \cdot e^{F\delta/k_BT}$. Using our data and the previously published data (***Larson et al., 2012***) collected at saturating [NTP] (1 mM) and various forces (***Figure 6***), we fit the $V_{max}$–$F$ dependence to ***Equation 16*** and obtained the values of $k_1 = 87 ± 61$ s$^{-1}$, $k_3 = 33 ± 8$ s$^{-1}$, and $\delta = 0.64 ± 0.58$ bp for the wild-type Pol II, and $k_1 = 65 ± 37$ s$^{-1}$, $k_3 = 62 ± 32$ s$^{-1}$, and $\delta = 0.64 ± 0.50$ bp for the mutant Pol II. Thus, the same qualitative conclusion that both translocation and catalysis are rate-limiting for the elongation reaction can be drawn from this alternative approach. Compared to the approach of using the nucleosomal barrier as a tool to determine $k_1$ and $k_3$, fitting the $V_{max}$–$F$ relationship involves one additional free parameter ($\delta$) and the values are less constrained (larger errors). In the future, it is worthwhile to use either of these two approaches or both to test whether prokaryotic transcription also employs a linear ratchet mechanism.

## The energy landscape for transcription elongation

With the same transcript length, RNAP is able to move back and forth on the DNA template, forming different TEC configurations (***Figure 7A,B***). Each translocation state corresponds to a local energy minimum (***Yager and von Hippel, 1991***; ***Bai et al., 2004***; ***Tadigotla et al., 2006***). Transitions between the pre- and post-translocated states, together with NTP binding and catalysis, constitute the active elongation pathway (***Figure 7—figure supplement 1***, green). The enzyme can also enter the pausing pathway by transiting from the pre-translocated state to the backtracked states (***Figure 7—figure supplement 1***, blue). The hyper-translocated states, in which the enzyme undergoes further forward translocation beyond 1 bp, are energetically unfavorable. The rate constants extracted from our single-molecule experiments translate into a free energy landscape for Pol II's mechanical translocations and chemical transitions (***Figure 7B***, ***Figure 7—figure supplement 1***; 'Materials and methods'), which reveals many detailed features of the kinetics of Pol II transcription.

First, the staircase shape formed by the energy minima of post-translocated, pre-translocated, and 1-bp backtracked states shows that the off-pathway backtracked states are thermodynamically more stable than the on-pathway states (***Figure 7B***). This feature confers the enzyme its propensity to enter

the pausing pathway, which is the central mechanism for various types of transcriptional control, such as arrest, proofreading, co-transcriptional RNA folding, and recruitment of regulators.

Second, the energy barrier from the pre-translocated to the 1-bp backtracked state is 2.5 $k_B$T higher than the barrier from the pre-translocated to the post-translocated state, causing $k_1$ to be more than 10 times faster than $k_{b1}$. Thus, at the beginning of each nucleotide addition cycle, the pre-translocated TEC favors the catalysis-competent post-translocated state kinetically over the 1-bp backtracked state, even though it is thermodynamically more favorable to move in the opposite direction. This property ensures that pausing only occurs sporadically so that the transcript can be synthesized within a reasonable amount of time. In addition, the barriers between neighboring backtracked states are also relatively high, preventing the enzyme from backtracking too far, which could lead to transcriptional arrest.

Third, the first backtracking step appears to be unique from further backtracking steps in two aspects. Kinetically, entering the 1-bp backtracked state is easier than entering subsequent backtracked states, as reflected by the difference between $k_{b1}$ and $k_{bn}$ (n≥2). Such a difference is supported by structural data: the structure of an arrested Pol II complex suggests that backtracking beyond 1 bp is disfavored as it is sterically hindered by a 'gating' tyrosine (Rpb2-Y769) (*Cheung and Cramer, 2011*). Thermodynamically, transiting from the pre-translocated state to the 1-bp backtracked state is favorable, while backtracking for more steps yields no additional energetic benefit. This result can also find structural support: the first backtracked nucleotide is stabilized by a binding pocket formed by several Pol II residues, whereas the second or third backtracked nucleotide makes no additional contact to the enzyme (*Wang et al., 2009*).

Thus, our model depicts an enzyme with a delicate balance between active elongation and inactive pausing (*von Hippel and Pasman, 2002*). This model can serve as a framework to study the effects of DNA sequence and nascent RNA structure on transcriptional dynamics (*Bai et al., 2004*; *Tadigotla et al., 2006*; *Zamft et al., 2012*). Moreover, this model may improve our understanding of the control of transcription fidelity. The 1-bp backtracked state is closely associated with the proofreading process of Pol II, as the enzyme in this location preferentially cleaves the 3′ dinucleotide of the RNA containing the mismatched base, empting the active site for NTP binding (*Wang et al., 2009*). It is possible that nucleotide misincorporation slows down forward translocation, thereby promoting the entry to the pausing pathway and the removal of the dinucleotide.

It is worth noting that we cannot definitively rule out the alternative scenario in which the first unique pausing state corresponds to a non-backtracked intermediate. Nonetheless, no evidence has been found for the universal occurrence of such an intermediate in Pol II transcription. The interpretation that most pauses in Pol II transcription are caused by enzyme backtracking is more parsimonious, especially given the corroborating structural data mentioned above.

## Roles of the TL element in transcriptional regulation

The kinetic characterization of the E1103G mutant Pol II reveals that this TL mutation results in many modifications to the enzyme dynamics (*Table 3*). Between the pre-translocated state and the post-translocated state, the mutant is significantly more biased toward the former than the wild type (*Figure 7B*). This property, together with its lower forward translocation rate, renders the mutant's elongation velocity more sensitive to perturbations of its forward translocation, such as an externally applied force (*Figure 6B*) or the presence of a nucleosomal barrier (*Figure 3B*). It has been shown that the inter-conversion between pre- and post-translocated states involves the transitions of the TL between an open conformation and a wedged conformation (*Brueckner and Cramer, 2008*). It is plausible that the mutation modulates the enzyme's translocation kinetics by altering the rates of transition between these two conformations.

Furthermore, our analyses lead to the conclusion that the faster overall elongation velocity of the mutant is due to its much greater catalysis rate despite a slower translocation step. The increase of the catalysis rate is most likely due to a faster NTP sequestration step induced by the closure of the TL (*Kireeva et al., 2008*). The lack of hydrogen bonding between T1095 and the mutated E1103 residue may destabilize the open state of the TL and speed up its closure (*Walmacq et al., 2012*).

The E1103G mutation also affects the pausing kinetics. Specifically, a decrease in the activation energy required to return from the first backtracked state to the pre-translocated state accelerates the recovery from a pause (*Figure 7B*). Consequently, the mutant populates the 1-bp backtracked state less than the wild-type. This property might affect the overall fidelity of transcription. It has been

previously shown that E1103G mutation strongly promotes incorporation of non-cognate NMP and mismatch extension (*Kaplan et al., 2008*; *Kireeva et al., 2008*). The destabilization of the 1-bp back-tracked state relative to the pre-translocated state in the E1103G mutant, established in this work, is consistent with its efficient mismatch extension and suggests that this mutation might also confer a defect in proofreading activity.

Together, our results suggest that the dynamics of TL are involved in multiple phases of transcription elongation, including translocation, catalysis, and pausing. In vivo, various transcription factors and small molecules can directly manipulate the TL dynamics and regulate transcription elongation. For example, transcription factor IIS (TFIIS) stimulates the endonuclease activity of Pol II by replacing the TL with its zinc finger domain, and thus, rescues transcription elongation by creating a new 3'-end of the transcript at Pol II's active site (*Kettenberger et al., 2003*). In fact, the viability of yeast cells expressing only the E1103G mutant Pol II is strictly dependent on TFIIS (*Malagon et al., 2006*). It is interesting to investigate how these trans-acting factors modify the rate-limiting mechanism and detailed kinetics of the elongation reaction. Finally, the elementary rate constants extracted from our analyses should provide a reference frame for future computational studies aiming to fully describe the molecular trajectory of a transcribing polymerase.

## Materials and methods

### Proteins and DNA preparation

Biotinylated wild-type and E1103G *S. cerevisae* Pol II (unphosphorylated C-terminal domain) were purified as previously described (*Kireeva et al., 2005*). The 3-kb DNA handle was prepared by PCR from Lambda DNA (NEB, Ipswich, MA) using a digoxigenin-labeled primer. The 574-bp DNA template was prepared by PCR from a modified pUC19 plasmid (*Zhang et al., 2006*) containing the 601 nucleosome positioning sequence (NPS) (*Lowary and Widom, 1998*). Each histone protein was recombinantly expressed and purified from *E. coli*, reconstituted to octamers (*Wittmeyer et al., 2004*), and loaded on the NPS-containing DNA using salt gradient dialysis (*Thåström et al., 2004*).

### Assembly of transcription elongation complexes

The transcription elongation complexes (TECs) were assembled by annealing a 9-nt RNA primer (IDT, Coralville, IA) to a 93-nt template DNA, incubating the hybrid with a biotinylated Pol II, and subsequently annealing a 96-nt complementary DNA using previously published sequences and procedures (*Hodges et al., 2009*). The TEC was walked to a stall site by addition of ATP, CTP and GTP. In the assisting force geometry, the downstream end of the stalled TEC was ligated to the 574-bp DNA containing the 601 NPS (with or without a preloaded nucleosome), while its upstream end was ligated to the 3-kb DNA handle. In the opposing force geometry, the downstream end of the TEC was ligated to a 4-kb DNA amplified from Lambda DNA (*Zamft et al., 2012*). The complexes were incubated with 2.1-μm streptavidin-coated beads (Spherotech, Lake Forest, IL), and DNA tethers were formed in a dual-trap optical tweezers instrument by attaching the digoxigenin-labeled DNA handle to a 2.1-μm anti-digoxigenin IgG-coated bead. In the assisting force geometry, Pol II and its upstream DNA were under tension, while no external force was applied to the downstream nucleosome (*Figure 2A*). The tension in the upstream DNA prevented intra-nucleosomal loop transfer and thus ensured that the nucleosome was always ahead of the transcribing polymerase. Transcription was restarted in optical tweezers by addition of NTPs (Thermo Fisher Scientific, Waltham, MA). The transcription buffer contains 20 mM Tris-HCl (pH 7.9), 5 mM $MgCl_2$, 10 μM $ZnCl_2$, 1 mM β-mercaptoethanol, 1 μM pyrophosphate, 300 mM KCl, and NTPs ranging from 35 μM to 2 mM each.

### Data collection and analysis

Position data were recorded at 2 kHz, averaged and decimated to 50 Hz, and filtered using a second-order Savitzky-Golay filter with a time constant of 1 s. The contour length of the DNA was calculated from the extension and force using the worm-like-chain formula of DNA elasticity (*Bustamante et al., 1994*) with a persistent length of 30 nm. This value of persistent length was obtained from pulling 3-kb DNA in our transcription buffer (data not shown). To alleviate calibration error and improve positional accuracy, single-molecule transcription traces that passed 85% of the template were aligned using both the stall site and the expected run-off length (*Bintu et al., 2012*). Shorter traces were also proportionally extended based on the average error from the run-off traces. To identify pauses, we computed the dwell time of Pol II at each nucleotide position. Pauses were identified from dwell times that were

longer the average dwell time by at least a factor of two. Due to the limited spatial resolution, we joined pauses that were separated by 3 bp or fewer into a single continuous pause. Pauses longer than 1 s are most likely caused by backtracking (*Maoileidigh et al., 2011*) and were counted. Pause-free velocities were calculated from time derivatives of the filtered position data, with a threshold of 2 nt/s to remove pauses. All curve fittings were performed by non-linear regression of the means weighted by the inverse of the variance.

## Monte Carlo simulation

From an elongation-competent state, Pol II can either elongate by 1 nt with the net forward translocation rate $k_1^{net}$ and incorporate an NMP to the RNA transcript, or enter a backtracked pause by 1 nt. During a pause, Pol II diffuses forward and backward with force-biased rate constants $k_f$ and $k_b$, respectively. For each condition, we simulated 100 trajectories and extracted the pause durations and densities to compare with the experimentally measured values.

## Estimation of the timescale of local nucleosomal DNA fluctuations

Fluorescence correlation spectroscopy and fluorescence resonance energy transfer experiments showed that the first 20–30 bp of DNA at the nucleosome ends spontaneously unwrap and rewrap on the histone surface every 10–250 ms (*Li et al., 2005*; *Koopmans et al., 2008*). The timescale of the 1-bp DNA fluctuations has not been directly reported but can be estimated from the experimental results above for longer DNA fluctuations. Assuming the wrapping/unwrapping kinetics is uniform along the DNA, we can model the unwrapping of a 25-bp DNA segment as:

$$0 \underset{k_w}{\overset{k_u}{\rightleftharpoons}} 1 \underset{k_w}{\overset{k_u}{\rightleftharpoons}} 2 \underset{k_w}{\overset{k_u}{\rightleftharpoons}} 3 ... \underset{k_w}{\overset{k_u}{\rightleftharpoons}} 24 \overset{k_u}{\rightarrow} 25$$

where $k_u$ and $k_w$ are the local unwrapping and wrapping rate constants of each basepair, respectively. Since the local wrapping equilibrium constant has been shown to be close to 1 (*Hodges et al., 2009*; *Bintu et al., 2012*), we further approximate $k_u$ and $k_w$ with a single value $k$:

$$0 \underset{k}{\overset{k}{\rightleftharpoons}} 1 \underset{k}{\overset{k}{\rightleftharpoons}} 2 \underset{k}{\overset{k}{\rightleftharpoons}} 3 ... \underset{k}{\overset{k}{\rightleftharpoons}} 24 \overset{k}{\rightarrow} 25$$

A net rate constant can substitute for each pair of forward and reverse rate constants (*Cleland, 1975*):

$$0 \xrightarrow{k_{0\to1}^{net}} 1 \xrightarrow{k_{1\to2}^{net}} 2 \xrightarrow{k_{2\to3}^{net}} 3 ... \xrightarrow{k_{23\to24}^{net}} 24 \xrightarrow{k_{24\to25}^{net}} 25$$

The net rate constants are given by:

$$k_{24\to25}^{net} = k$$

$$k_{23\to24}^{net} = k \cdot \frac{k_{24\to25}^{net}}{k_{24\to25}^{net} + k} = \frac{k}{2}$$

$$k_{22\to23}^{net} = k \cdot \frac{k_{23\to24}^{net}}{k_{23\to24}^{net} + k} = \frac{k}{3}$$

$$\vdots$$

$$k_{1\to2}^{net} = k \cdot \frac{k_{2\to3}^{net}}{k_{2\to3}^{net} + k} = \frac{k}{24}$$

$$k_{0\to1}^{net} = k \cdot \frac{k_{1\to2}^{net}}{k_{1\to2}^{net} + k} = \frac{k}{25}$$

The time required for unwrapping 25 bp of DNA equals the total time of unwrapping each bp of DNA:

$$\tau_{0\to25} = \frac{1}{k_{0\to1}^{net}} + \frac{1}{k_{1\to2}^{net}} + ... + \frac{1}{k_{24\to25}^{net}} = \frac{325}{k} \approx (10-250) ms$$

Thus, the time for 1-bp DNA to unwrap from the nucleosome is expected to be less than 1 ms:

$$\tau_{0\rightarrow1} = \frac{1}{k} < 1\,ms$$

In the same way, we can also show that the 1-bp DNA rewrapping occurs on a similar timescale ($\tau_{1\rightarrow0} < 1$ ms). In addition, molecular dynamics simulations also suggested that the local nucleosomal DNA fluctuates very fast (ns–µs timescale) (*Voltz et al., 2012*). Therefore, we assume that the 1 bp of DNA in front of the polymerase unwraps and rewraps much faster than the translocation of the enzyme.

## Correction for undercounted short pauses

Experimentally we only counted pauses with lifetimes between 1 s and 120 s. The total pause density $\rho_{pause,\,total}$ is given by:

$$\rho_{pause,total} = \frac{k_b}{k_b + k_1^{net}} = \frac{\rho_{pause,1<t<120}}{\int_1^{120} \psi(t)dt}$$

The correction factor can be solved analytically to be 2.9 for the wild-type Pol II. For the mutant enzyme, the values of $k_{f1}$ and $k_{b1}$ are different from those for the wild-type (*Table 3*). We simulated transcriptional pauses using the lower bounds of $k_{f1}$ and $k_{b1}$ and obtained a correction factor of ~7 for the mutant Pol II.

## Estimation of the value of K for the mutant Pol II

The value of $K$ for the mutant Pol II ($K_{mutant}$) cannot be constrained by *Equation 14* due to the relatively large experimental error. We took a different approach to constrain $K_{mutant}$ by simulating the $\rho_{pause}$–[NTP] relationship with varying $K_{mutant}$ values and then comparing it to the experimental data (*Figure 5—figure supplement 1*). We found that the simulated curve substantially deviates from the experimental curve when $K_{mutant}$ becomes larger than 100 µM. Hence, we set the upper bound of $K_{mutant}$ to be 100 µM. Using the $k_{-1}K$ value of $(2.5 \pm 0.4) \times 10^4$ µM·s$^{-1}$ obtained from *Equation 13*, we set the lower bound of $k_{-1}$ for the mutant to be 210 s$^{-1}$. The notion that the NTP dissociation rate is much faster than the catalysis rate ($k_{-2} >> k_3$) has been widely used in the kinetic studies of RNA and DNA polymerases, and is supported by biochemical evidence (*Rhodes and Chamberlin, 1974*; *Johnson, 1993*; *Foster et al., 2001*; *Bai et al., 2004*; *Maoiléidigh et al., 2011*). It follows from this notion that $K = (k_{-2}+k_3)/k_2 \approx k_{-2}/k_2$. Thus, $K$ becomes virtually identical to $K_D$, the NTP dissociation constant. Because the mutated residue (Glu1103) is located distal from the NTP-interacting part of TL (*Wang et al., 2006*) and the E1103G mutation affects TL closure and NTP sequestration after the initial docking step (*Kireeva et al., 2008*), the NTP binding/dissociation kinetics are unlikely to be markedly affected by the mutation. Therefore, it is reasonable to assume that the wild-type and the mutant enzymes share similar $K_D$ values (~9.2 µM). Under this assumption, we could estimate the $k_{-1}$ value for the mutant to be ~2700 s$^{-1}$.

## Construction of the energy landscape

The free energy difference ($\Delta\Delta G$) between two neighboring translocation states was computed using the forward and reverse rate constants between these states ($k_+$ and $k_-$):

$$\Delta\Delta G = -k_B T \cdot \ln\left(\frac{k_-}{k_+}\right)$$

The height of the activation energy barrier ($\Delta G^\dagger$) was calculated according to the Arrhenius equation:

$$k = A \cdot \exp\left(-\Delta G^\dagger / k_B T\right)$$

where $k$ is the corresponding rate constant and $A$ is the pre-exponential factor. In this study, for illustration purposes, we made a simplifying assumption that all reaction steps share the same pre-exponential factor. $A$ was calculated using the stepping rate constant during backtracking $k_0$ = 1.3 s$^{-1}$ and the barrier height between neighboring backtracked states $\Delta G_b^\dagger$. We assumed that one DNA–RNA hybrid basepair and one DNA–DNA basepair in the transcription bubble must be broken before any other bonds are formed and that no other interactions contribute to the barrier (*Tadigotla et al., 2006*). Using the available free energy data for basepairing (*Sugimoto et al., 1996*; *Wu et al., 2002*), we estimated $\Delta G_b^\dagger$ to be ~8.5 $k_B$T, which translates to an Arrhenius pre-factor of ~6.4 × 10$^3$ s$^{-1}$. The catalysis step is essentially irreversible in our experimental condition. The free energy drop after each nucleotide addition cycle is arbitrarily set to be 10 $k_B$T.

## Acknowledgements

We thank Lacramioara Bintu, Gheorghe Chistol, Yves Coello, Craig L Hetherington, Masahiko Imashimizu, Troy A Lionberger, Ninning Liu, Yara X Mejia, and Maya Sen for critical readings of the manuscript, and W Gregory Alvord for help with statistical analyses. We would like to dedicate this work to the memory of W Wallace Cleland whose pioneering studies on enzyme kinetics provide the foundation for this work.

## Additional information

### Funding

| Funder | Grant reference number | Author |
| --- | --- | --- |
| National Institutes of Health | R01-GM032543 | Carlos J Bustamante |
| Department of Energy | DE-AC02-05CH11231 | Carlos J Bustamante |
| Howard Hughes Medical Institute | | Carlos J Bustamante |

The funders had no role in study design, data collection and interpretation, or the decision to submit the work for publication.

### Author contributions

MD, TI, Conception and design, Acquisition of data, Analysis and interpretation of data, Drafting or revising the article; SL, CJB, Conception and design, Analysis and interpretation of data, Drafting or revising the article; MLK, MK, Analysis and interpretation of data, Drafting or revising the article, Contributed unpublished essential data or reagents; LL, Drafting or revising the article, Contributed unpublished essential data or reagents

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
