## [Decision Letter]

Thank you for sending your work entitled “Complete dissection of transcription elongation reveals slow translocation of Pol II in a linear ratchet mechanism” for consideration at *eLife*. Your article has been favorably evaluated by a Senior editor and 3 reviewers, one of whom is a member of our Board of Reviewing Editors.

The Reviewing editor and the other reviewers discussed their comments before we reached this decision, and the Reviewing editor has assembled the following comments to help you prepare a revised submission.

All three reviewers find your work of significant interest and high quality. By measuring the transcription kinetics on bare and nucleosomal DNA, this single-molecule force study dissected the transcription elongation pathway, determined each individual rate constant on the pathway, and mapped out the energy landscape for transcription elongation and backtracking. Instead of assuming that the reversible transition between pre- and post-translocated states of the nucleotide addition cycle occurs much faster than the subsequent catalysis steps, this study found that the forward translocation rate is comparable to the catalysis rate. The results suggest that a linear Brownian Ratchet Model can explain the measured behavior of transcription elongation kinetics. The reviewers feel that the findings in this work are important, the experimental design is elegant, and that the data are of high quality.

The reviewers have the following concerns regarding the use of the nucleosome barrier as a tool to separately measure the forward and reverse translocation rates. Pol II interactions with a nucleosome are rather complex and hence could complicate the interpretation of data here. One of the key assumptions is that the local nucleosomal DNA fluctuations (wrapping and unwrapping) are faster than Pol II forward translocation. Is this assumption valid for the present study, which uses nucleosomes with the 601 sequence and involves predominant transcriptional pauses close to the nucleosomal dyad? What is the evidence that wrapping/unwrapping kinetics near the dyad of the 601 nucleosome is faster than Pol II translocation? How do the different interaction strengths at different sites of the nucleosome affect the results in this work? Is it possible to obtain the same conclusion using a simpler system without involving nucleosomes? For example, the application of force can also perturb the energy landscape. *k*_1_, *k*_3_, and δ may be obtained by examining *V*_max_ as a function of force. This approach removes the need for [Disp-formula equ6] and may potentially remove the need for using nucleosomes. Given the importance of the nucleosome barrier as an experimental tool in the work, the advantages and requirements of using this system should be more clearly explained and justified in the main text.

The reviewers also feel that the manuscript is written in a very concise and dense manner. It is not easy to follow, especially for non-experts. Give that *eLife* has a broad audience, we suggest that you expand the manuscript to make it easier to read and understand. In particular, the derivations of various kinetic rate constants from experimentally measured quantities are nontrivial and the manuscript often relied on citations of previous papers rather than explaining clearly the rationale of how these rate constants are derived. The manuscript should be made more self-sufficient.

---

## [Author Response]

*The reviewers have the following concerns regarding the use of the nucleosome barrier as a tool to separately measure the forward and reverse translocation rates. Pol II interactions with a nucleosome are rather complex and hence could complicate the interpretation of data here. One of the key assumptions is that the local nucleosomal DNA fluctuations (wrapping and unwrapping) are faster than Pol II forward translocation. Is this assumption valid for the present study, which uses nucleosomes with the 601 sequence and involves predominant transcriptional pauses close to the nucleosomal dyad? What is the evidence that wrapping/unwrapping kinetics near the dyad of the 601 nucleosome is faster than Pol II translocation? How do the different interaction strengths at different sites of the nucleosome affect the results in this work? Is it possible to obtain the same conclusion using a simpler system without involving nucleosomes? For example, the application of force can also perturb the energy landscape.* k_*1*_*,* k_*3*_*, and δ may be obtained by examining* V_*max*_
*as a function of force. This approach removes the need for*
[Disp-formula equ6]
*and may potentially remove the need for using nucleosomes. Given the importance of the nucleosome barrier as an experimental tool in the work, the advantages and requirements of using this system should be more clearly explained and justified in the main text*.

The reviewers correctly pointed out that the derivation of *k*_1_ and *k*_3_ relies on the assumption that nucleosomal DNA fluctuations are much faster than Pol II forward translocation. The timescale of DNA fluctuations at the nucleosome ends has been studied. Fluorescence correlation spectroscopy and fluorescence resonance energy transfer experiments showed that the first 20–30 bp of nucleosomal DNA spontaneously unwrap and rewrap every 10–250 ms (Li et al., NSMB 2005; Koopmans et al., Chemphyschem. 2008). Since the polymerase moves 1 bp per step, the timescale of Pol II translocation (10 ms–1 s) should be compared to that of one-bp DNA fluctuation, which has not been directly reported but can be estimated from the above experimental results. The average time for 1-bp DNA to open or close on the histone surface is expected to be less than 1 ms (a derivation is shown in the Materials and methods section). In addition, coarse-grained molecular dynamics simulations showed that the first 9-bp DNA segment fluctuates very fast (ns–µs timescale) (Voltz et al., Biophys. J. 2012). Therefore, the local DNA in front of the polymerase most likely fluctuates much faster than the translocation of the polymerase. The timescale of DNA fluctuations near the nucleosomal dyad has not been directly studied. Nonetheless, we have previously compared the local wrapping/unwrapping equilibrium constants between the entry and the central regions of the nucleosome and found that they are very similar (Bintu et al., Cell 2012). It is difficult to envision a scenario in which both wrapping and unwrapping rates would slow down by the exact same amount in the central region compared to the entry region of the nucleosome. A much more likely scenario is that the wrapping/unwrapping kinetics of DNA in front of a transcribing polymerase is relatively uniform across the whole nucleosome region. Therefore, we think that the assumption of fast DNA fluctuation compared to Pol II translocation is a reasonable one.

Regarding the predominant pauses observed near the nucleosomal dyad with the 601 sequence, we have shown that they are due to the lack of secondary structures formed in the nascent transcript behind the transcribing enzyme in this region that induces extensive Pol II backtracking, rather than due to particularly strong histone-DNA interactions (Bintu et al., Cell 2012). Moreover, the pauses were also observed on bare DNA with the same sequence, although to a lesser degree (Figure 2—figure supplement 4). In addition, gel-based assays showed a lack of 5-bp or 10-bp pausing periodicity within the nucleosome region for yeast Pol II (Kireeva et al., Mol. Cell 2005; Bondarenko et al., Mol. Cell 2006). Such periodicity would be expected if the strength of the histone-DNA interactions were the major determinant for Pol II pausing (Luger et al., Nature 1997; Hall et al., NSMB 2009). Therefore, we do not think that the strong pauses near the dyad drastically affect the on-pathway kinetic parameters of Pol II (i.e., *k*_1_ and *k*_3_), which were derived from the pause-free velocity. To further rule out the possibility that the different behavior of Pol II near the dyad may bias the values of these kinetic rates, we reanalyzed the pause-free velocity data without a 40-nt span (−35 to +5 relative to the dyad) in the central region of the nucleosome and obtained *k*_1_ = 158±43 s^-1^ and *k*_3_= 33±2 s^-1^ for the wild-type Pol II, and *k*_1_ = 59±4 s^-1^ and *k*_3_ = 120±13 s^-1^ for the mutant Pol II at 6.5 pN of applied aiding force. These values are close to those obtained with all the pause-free velocity data and lead to the same conclusion that *k*_1_ and *k*_3_ are of the same order of magnitude.

Nonetheless, it is true that the interaction between Pol II and the nucleosome is rather complex and their interaction may vary at different sites. We simplified the problem by using the averaged values of the pause-free velocity and the fraction of time the local nucleosomal DNA is unwrapped (γ_u_) over the whole nucleosome region to extract the intrinsic kinetic properties of Pol II. This point was probably not stressed enough in the original manuscript. It is now clearly stated in the revised manuscript.

The reviewers also made an excellent suggestion that *k*_1_, *k*_3_, and *δ* may be extracted by examining *V*_max_ as a function of force. This is indeed possible: *V*_max_(*F*) = *k*_1_(*F*)*k*_3_/(*k*_1_(*F*)+*k*_3_)**d*, where *k*_1_(*F*) = *k*_1_(0)*e*^*Fδ/k_B_*^^*T*^. Using our data and the published data from Larson et al. (Larson et al., PNAS 2012) collected at saturating [NTP] (1 mM) and various forces (Figure 6), we fit the *V*_max_-*F* dependence to the above equation and obtained *k*_1_= 87±61 s^-1^, *k*_3_= 33±8 s^-1^, and *δ* = 0.64±0.58 bp for the wild-type Pol II, and *k*_1_= 65±37 s^-1^, *k*_3_= 62±32 s^-1^, and *δ* = 0.64±0.50 bp for the mutant Pol II. Thus, the same qualitative conclusion, i.e., *k*_1_ and *k*_3_ are comparable and translocation is one of the rate-limiting steps, can be drawn from this analysis. Compared to the approach in which the nucleosomal barrier is used to specifically perturb forward translocation, fitting the *V*_max_-*F* dependence involves one additional free parameter (*δ*) and the values are less constrained. If we fix *δ* at 0.5 bp and only fit for *k*_1_ and *k*_3_ using the *V*_max_-*F* dependence, the values are better bound (*k*_1_= 72±13 s^-1^ and *k*_3_= 35±3 s^-1^ for the wild-type; *k*_1_= 55±10 s^-1^ and *k*_3_= 75±13 s^-1^ for the mutant). However, this analysis involves one additional assumption (*δ* = 0.5 bp), which may not always be valid as is the case for the mutant Pol II (*δ* = 0.24±0.05 bp). One reason for this relatively poor fitting is that force has a relatively small effect on *k*_1_, given the small step size of the polymerase (1 bp or 0.34 nm). The nucleosomal barrier effectively decreases *k*_1_ by 40%. The same amount of reduction would require 13 pN of opposing force, which is not feasible because the operational stall force for Pol II is only ∼8 pN due to backtracking (Galburt et al., Nature 2007). In the revised Discussion section, we have added a comparison between the two approaches of using force or nucleosome to extract the kinetic parameters of Pol II transcription.

*The reviewers also feel that the manuscript is written in a very concise and dense manner. It is not easy to follow, especially for non-experts. Give that* eLife *has a broad audience, we suggest that you expand the manuscript to make it easier to read and understand. In particular, the derivations of various kinetic rate constants from experimentally measured quantities are nontrivial and the manuscript often relied on citations of previous papers rather than explaining clearly the rationale of how these rate constants are derived. The manuscript should be made more self-sufficient*.

In the revised Results section, we have included the derivations of various rate constants as well as the corresponding rationales. We have also expanded the text considerably to more clearly explain the results and their implications.